# Preoperative Serum GDF-15, Endothelin-1 Levels, and Intraoperative Factors as Short-Term Operative Risks for Patients Undergoing Cardiovascular Surgery

**DOI:** 10.3390/jcm10091960

**Published:** 2021-05-02

**Authors:** Takashi Kato, Toshiaki Nakajima, Taira Fukuda, Ikuko Shibasaki, Takaaki Hasegawa, Koji Ogata, Hironaga Ogawa, Shotaro Hirota, Hirotaka Ohashi, Shunsuke Saito, Yusuke Takei, Masahiro Tezuka, Masahiro Seki, Toshiyuki Kuwata, Masashi Sakuma, Shichiro Abe, Shigeru Toyoda, Teruo Inoue, Hirotsugu Fukuda

**Affiliations:** 1Department of Cardiovascular Surgery, Dokkyo Medical University, Tochigi 321-0293, Japan; takato@dokkyomed.ac.jp (T.K.); sibasaki@dokkyomed.ac.jp (I.S.); kooga@dokkyomed.ac.jp (K.O.); hironaga_0722@yahoo.co.jp (H.O.); hirota-s@dokkyomed.ac.jp (S.H.); oh-1212@dokkyomed.ac.jp (H.O.); saitos@dokkyomed.ac.jp (S.S.); y-takei@dokkyomed.ac.jp (Y.T.); mtezuka@dokkyomed.ac.jp (M.T.); seki-m@dokkyomed.ac.jp (M.S.); tkuwata@dokkyomed.ac.jp (T.K.); fukuda-h@dokkyomed.ac.jp (H.F.); 2Department of Cardiovascular Medicine, Dokkyo Medical University, Tochigi 321-0293, Japan; masakuma@dokkyomed.ac.jp (M.S.); abenana@dokkyomed.ac.jp (S.A.); s-toyoda@dokkyomed.ac.jp (S.T.); inouet@dokkyomed.ac.jp (T.I.); 3Department of Medical KAATSU Training, Dokkyo Medical University, Tochigi 321-0293, Japan; thasegawa6134@gmail.com; 4Department of Liberal Arts and Human Development, Kanagawa University of Human Services, Kanagawa 238-8522, Japan; fukuda-h9w@kuhs.ac.jp

**Keywords:** growth differentiation factor-15, GDF-15, endothelin-1, ET-1, cardiovascular surgery, STS risk score, operative risk, acute kidney injury, mortality, morbidity, cardiopulmonary bypass time

## Abstract

Objectives: The Society of Thoracic Surgeons (STS) risk score is widely used for the risk assessment of cardiac surgery. Serum biomarkers such as growth differentiation factor-15 (GDF-15) and endothelin-1 (ET-1) are also used to evaluate risk. We investigated the relationships between preoperative serum GDF-15, ET-1 levels, and intraoperative factors and short-term operative risks including acute kidney injury (AKI) for patients undergoing cardiovascular surgery. Methods: In total, 145 patients were included in this study (92 males and 53 females, age 68.4 ± 13.2 years). The preoperative STS score was determined, and the serum GDF-15 and ET-1 levels were measured by ELISA. These were related to postoperative risks, including AKI, defined according to the Acute Kidney Injury Network (AKIN) classification criteria. Results: AKI developed in 23% of patients. The GDF-15 and ET-1 levels correlated with the STS score. The STS score and GDF-15 and ET-1 levels all correlated with preoperative eGFR, Alb, Hb, and BNP levels; perioperative data (urine output); ICU stay period; and postoperative admission days. Patients with AKI had longer circulatory pulmonary bypass (CPB) time, and male patients with AKI had higher ET-1 levels than those without AKI. In multivariable logistic regression analysis, the preoperative ET-1 level and CPB time were the independent determinants of AKI, even adjusted by age, sex, and BMI. The preoperative GDF-15 level, CPB time, and RCC transfusion were independent determinants of 30-day mortality plus morbidity. Conclusion: Preoperative GDF-15 and ET-1 levels as well as intraoperative factors such as CPB time may be helpful to identify short-term operative risks for patients undergoing cardiovascular surgery.

## 1. Introduction

Cardiovascular surgery can be considered even for the elderly and high-risk patients thanks to technological advances, but postoperative complications and mortality remain serious problems. Scoring systems such as the Society of Thoracic Surgeons (STS) score are widely used to assess operative risks, including acute kidney injury (AKI) and perioperative outcomes [1,2]. These risk scores are based on a collected database and can be used to predict the risk of death from surgery by entering patient demographics and clinical variables into an easy-to-use online calculator. AKI is an important and frequent complication occurring during the postoperative period in patients undergoing cardiovascular surgery and is associated with increased morbidity, increased short- and long-term mortality, and poor prognosis [3,4,5]. Therefore, early preventive measures such as preoperative risk prediction and perioperative optimization are necessary. The risk model for AKI after cardiovascular surgery focuses primarily on the need for dialysis [6], and STS scores have been reported to predict the risk of postoperative dialysis in patients undergoing coronary artery bypass grafting (CABG) [7]. However, the STS score is not sufficient in terms of predicting mild to moderate postoperative AKI that does not lead to the introduction of dialysis.

The precise pathophysiology of AKI associated with cardiovascular surgery remains unsettled [8]. The known predictors of postoperative AKI include pre-existing renal dysfunction, peripheral arterial disease, diabetes, old age, female sex, congestive heart failure, and left ventricular ejection fraction (LVEF) disorders [9,10] as well as intraoperative data such as CPB duration [11,12,13]. Nonetheless, one of the challenges in preventing AKI is a lack of accurate early predictors. Serum creatinine is useful for early diagnosis of AKI, but its concentration is also affected by other non-renal factors, such as age, gender, muscle mass, and nutritional status [14]. Therefore, the renal function appears not to be completely reflected to predict AKI [15]. Biomarkers can also provide insights into underlying mechanisms and lead to better understanding of the pathophysiology of complex disease states such as AKI. This enhanced understanding can then be integrated into disease management, which can lead to better therapies and ultimately to improved patient outcomes. Now, biomarkers being studied for AKI prediction include neutrophil gelatinase-associated lipocalin, kidney injury molecule-1, and N-acetyl-β-D-glucosaminidase [16,17,18]. However, none of these markers have been shown to predict postoperative AKI in cardiovascular surgery patients when evaluated in preoperative settings.

Endothelin-1 (ET-1) is a 21-amino-acid peptide produced by numerous tissues, including the vascular endothelium and kidney [19]. Endothelin acts as a potent vasoconstrictor of cortical and medullary vessels, regulates renal function including glomerular filtration, and functions as a natriuretic peptide by inhibiting the reabsorption of sodium and water along the collecting duct. Thus, endothelin plays an important pathophysiological role in renal disorders characterized by increased vascular resistance, including renal ischemia [20,21]. Ghashghaei et al. [22] have reported that elevated preoperative ET-1 level poses an AKI risk after coronary artery bypass surgery. Growth differentiation factor-15 (GDF-15) is a cytokine associated with the transforming growth factor-β (TGF-β) superfamily [23]. Plasma levels of GDF-15 increase under pathological conditions such as hypoxia, inflammation, and oxidative stress and are closely associated with overall mortality [24]. GDF-15 is also secreted in the early stages of renal endothelial dysfunction, which involves microalbuminuria and renal impairment [25,26]. In addition, it has been reported to identify individuals at high risk of developing chronic kidney disease (CKD) and as an independent marker of mortality [27]. Recent studies have identified a strong association between GDF-15 levels and renal dysfunction [28,29], and it has been reported that increased preoperative GDF-15 levels can predict AKI risk after CABG [30]. However, the usefulness of these preoperative serum biomarkers (GDF-15 and ET-1) for predicting operative risks including AKI remains unsettled.

Therefore, the purpose of the present study is to investigate the relationships between preoperative serum GDF-15, ET-1 levels, and intraoperative factors and short-term operative risks including AKI for patients undergoing cardiovascular surgery.

## 2. Materials and Methods

### 2.1. Participants

Patients undergoing planned cardiovascular surgery with cardiopulmonary bypass (CPB) during thoracotomy or median sternotomy at Dokkyo Medical Hospital from October 2015 to December 2020 were included in this study. The Regional Ethics Committee of Dokkyo Medical University has approved the study protocol (approval number: 27077), which was conducted according to the Declaration of Helsinki. Each patient provided written consent. We prospectively investigated the onset of AKI and short-term risk after the operation. The end point and time of the follow-up was one month after the operation or death.

Fasting blood samples were obtained in tubes containing EDTA sodium and in polystyrene tubes without an anticoagulant. Plasma was immediately separated by centrifugation at 3000 rpm at 4 °C for 10 min, and serum was collected by centrifugation at 1000 rpm at room temperature for 10 min. Hemoglobin A1 (HbA1c), brain natriuretic peptide (BNP), creatinine, hemoglobin (Hb), albumin (Alb), and estimated glomerular filtration rate (eGFR) were measured before the operation. The biochemical data were analyzed using routine chemical methods in the Dokkyo Medical University Hospital clinical laboratory. Levels of the inflammatory marker, high-sensitivity C-reactive protein (hsCRP), were measured by a latex-enhanced nephelometric immunoassay (N Latex CRP II and N Latex SAA, Dade Behring Ltd., Tokyo, Japan).

To measure GDF-15 and ET-1 levels, blood samples were drawn into pyrogen-free tubes without EDTA on the morning of cardiovascular surgery. The serum was stored in aliquots at −80 °C for all enzyme-linked immunosorbent assays (ELISAs).

### 2.2. Enzyme-Linked Immunosorbent Assay (ELISA)

Serum GDF-15 level was measured by the Human Quantikine ELISA Kit (DGD150 for GDF-15, R&D Systems, Minneapolis, MN, USA) as previously described [29]. The detection threshold of GDF-15 was 2.0 pg/mL. The serum concentrations of ET-1 level were measured by endothelin-1 Quantikine ELISA Kit (DET100, R&D Systems, Inc., Minneapolis, MN), and the detection threshold was 0.087 pg/mL.

### 2.3. Classification of AKI Classification

To identify postoperative AKI, the criteria of Kidney Disease Improving Global Outcomes (KDIGO) [31] were used. AKI was diagnosed as a sudden decrease in renal function when at least one of these criteria was satisfied: 1) The serum creatinine level was increased by ≥0.3 mg/dL within 48 h, or 2) serum creatinine level was increased by ≥1.5 times from the basal creatinine within 7 days. We did not consider a sudden decrease in the urinary output to define AKI because, as suggested in the KDIGO guidelines, the use of urine output criteria for diagnosis and staging has been less well validated, and in individual patients, the need for clinical judgment regarding the effects of drugs such as diuretics, fluid balance, and other factors must be included as indicated previously [30].

### 2.4. Items to Consider

Preoperative factors include age, sex, height, weight, body surface area, BMI, presence of hypertension (HT), diabetes (DM), dyslipidemia (Dlp), smoking, and hemodialysis (HD). Additional items considered were history of open-chest surgery and preoperative data (serum ET-1 level, GDF-15 level, hsCRP, Alb, creatinine, eGFR, BNP, Hb, preoperative left ventricular ejection fraction (LVEF), and preoperative NYHA). At present, the STS risk score is the most widely used risk score in the United States to estimate mortality risk after cardiac surgery [1,2,32,33]. We used the updated STS short-term risk calculator to reflect the latest 2018 adult cardiac surgery risk models. We assessed intraoperative factors such as the surgical procedure, CPB time, intraoperative bleeding, intraoperative urine volume, intraoperative blood transfusion volume (RCC, red blood cell concentrates; FFP, fresh frozen plasma; PC, platelet concentrates), postoperative result, AKI incidence rate, AKI stage, and major complications. We also evaluated illness, postoperative intubation time, intensive care unit (ICU) stay period, postoperative admission days, 30-day mortality, and 30-day morbidity. Morbidity was defined as follows: renal failure (requiring new HD), permanent stroke, prolonged ventilation (>48 h), deep sternal wound infection (DSWI), and re-operation.

### 2.5. CPB Management

Management during CPB was based on the following items: 1) a perfusion flow rate of 2.0 to 2.5 L/min/m^2^, perfusion pressure of 60 to 80 mmHg, and Hb concentration of 7 g/dL or higher. In addition, the goal of the operation was that the mixed venous oxygen saturation (SVO_2_) in CPB should be higher than 70% mmHg. When the Hb concentration was lower than 7 g/dL, administration of RCC was considered. To obtain the target urine volume in CPB of at least 1.0 mL/kg/h, the perfusion flow rate and perfusion pressure were adjusted.

### 2.6. Statistical Analysis

All data are presented as mean ± SD. The comparison of means between groups was carried out using the Mann–Whitney U-test or Student *t*-test. After testing for normality examination (Kolmogorov–Smirnov test or Shapiro–Wilk test), the comparison of means between groups were analyzed by a two-sided, unpaired Student’s *t*-test in the case of normally distributed parameters or by the Mann–Whitney-U-test in the case of non-normally distributed parameters. Associations among parameters were evaluated using Pearson or Spearman correlation coefficients. ROC curves were plotted to identify an optimal cutoff level of the perioperative data (CPB time) to detect AKI. Multivariable logistic regression analysis with clinical outcome such as AKI as the dependent variable was performed to identify independent factors (preoperative data (eGFR, STS score, ET-1, and GDF-15) or perioperative data). To select independent factors in logistic regression analysis, differences in preoperative or perioperative factors including CPB time with clinical outcome were examined using the U- and *t*-tests. The variable reduction method with all the factors that gave a significant difference was used to determine the independent factors in logistic regression analysis. ET-1 and GDF15, a biomarker investigated in this study, were considered as independent factors from the clinical aspect in this study. Using the independent factors obtained from the above results, logistic regression analysis was performed using the forced input method. Age, sex, and BMI were employed as covariates. All analyses were performed using SPSS version 24 (IBM Corp., New York, USA) for Windows. A *p* value of 0.05 was regarded as significant.

## 3. Results

### 3.1. Correlation between GDF-15, ET-1, STS Score, and Preoperative Blood Data

The baseline characteristics and perioperative and postoperative complications of the patients are summarized in Table 1 and Table 2. There were 92 males and 53 females. Their age was 68.4 ± 13.2 years (mean ± SD), and body mass index (BMI) was 23.5 ± 4.1 kg/m^2^. Table 1 shows the co-incidence of conventional risk factors, such as HT, DM, Dlp, smoking, and HD, and surgical procedures.

The mean values of STS score and preoperative serum levels of GDF-15 and ET-1 were 3.6% ± 4.2%, 1851 ± 1638 pg/mL, and 1.38 ± 0.92 pg/mL, respectively. The eGFR was 56.6 ± 27.3 mL/min/1.73 m^2^, and the BNP level was 422 ± 659 pg/mL.

The correlations between serum GDF-15, ET-1 concentration, and STS score are shown in Appendix A. The preoperative GDF-15 and ET-1 levels were correlated with STS score (GDF-15, r = 0.545, *p <* 0.001, Figure 1Ab; ET-1, r = 0.425, *p <* 0.001, Figure 1Ac). The STS score was significantly positively correlated with age (r = 0.563, *p <* 0.001), hsCRP (r = 0.191, *p* = 0.032), creatinine (r = 0.211, *p* = 0.018), and BNP level (r = 0.542, *p <* 0.001, Figure 1Ca), while it was negatively correlated with eGFR (r = −0.454, *p <* 0.000, Figure 1Ba), Hb (r = −0.602, *p <* 0.001), and Alb (r = −0.497, *p <* 0.001). The preoperative GDF-15 level was significantly positively correlated with age (r = 0.330, *p <* 0.001), hsCRP (r = 0.308, *p <* 0.001), creatinine (r = 0.586, *p ≤* 0.001), and BNP level (r = 0.412, *p <* 0.001, Figure 1Cb). On the other hand, it was negatively correlated with BMI (r = −0.190, *p* = 0.025), eGFR (r = −0.664, *p <* 0.001 Figure 1Bb), Hb (r = −0.491, *p <* 0.001), Alb (r = −0.519, *p <* 0.001), and EF (r = −0.275, *p* = 0.002). The preoperative ET-1 level was significantly positively correlated with creatinine (r = 0.351, *p <* 0.001), and BNP level (r = 0.630, *p <* 0.001, Figure 1Cc). On the other hand, it was negatively correlated with eGFR (r = −0.402, *p <* 0.001, Figure 1Cb), Hb (r = −0.253, *p* = 0.002), Alb (r = −0.340, *p ≤* 0.001), and EF (r = −0.237, *p* = 0.006). Thus, preoperative GDF-15 and ET-1 levels appear to be markers of operative risk just as the STS score is in cardiovascular surgery patients.

### 3.2. Correlation between GDF-15, ET-1, STS Score, and Perioperative Data

The correlations between preoperative serum GDF-15 level, ET-1 level, and STS score and intraoperative data and short-term risks are shown in Table 3. The preoperative GDF-15 level and STS score were correlated with operative time. The preoperative GDF-15 level and STS score were positively correlated with RCC during the operation. The STS score, GDF-15 level, and ET-1 level correlated with urine output. The ET-1 level and STS score were correlated with the postoperative intubation period.

### 3.3. Differences of Various Clinical Parameters between Patients with and without AKI

The perioperative and postoperative complications are illustrated in Table 2. Appendix A summarizes the comparative data from patients with and without AKI. AKI developed in 29 out of 126 non-HD patients (23%). The incidence rate of AKI was 25 patients out of 77 (32.5%) for males and 4 patients out of 49 (8%) for females. Thus, the incidence of AKI in females was considerably less than in males in our study. Patients with AKI had low BMI (*p* < 0.05), long CPB time (*p* < 0.001), high bleeding (*p* < 0.001), high RCC (*p* < 0.01), high FFP (*p* < 0.01), high PC, long postoperative intubation time (*p* < 0.001), and long ICU stay period (*p* < 0.001) in total patients and in males (Appendix A). The preoperative ET-1 level in males was significantly higher in patients with AKI than that in those without AKI. However, the preoperative GDF-15 level and STS score were not significantly different between patients with and without AKI. Thus, the development of AKI appears to be correlated with perioperative data (CPB time, bleeding, RCC, FFP, and postoperative intubation time) in total patients and preoperative ET-1 level in males. On the other hand, as shown in Table 2, the number of patients with stage 2 AKI was only 5, and the number of patients with stage 3 AKI was only 4. We have not obtained the statistical difference between AKI stage and STS score, ET-1, and GDF-15 level (data not shown).

Multivariable logistic regression analysis with AKI as the dependent variable was performed to identify independent factors (preoperative and intraoperative data). First, we investigated intraoperative data (CPB time, bleeding, RCC, FFP, PC, and postoperative incubation time) as independent factors, as shown in Table 4A. In multivariable logistic regression analysis that adjusted for age, gender, and BMI, CPB time remained significantly associated with the risk of developing AKI. Furthermore, multivariable logistic regression analysis with AKI as the dependent variable was performed to identify independent factors (preoperative and intraoperative data), as shown in Table 4B. The risk of developing AKI increased with increasing preoperative ET-1 level at baseline (OR for 1 pg/mL increase in ET-1, 2.177; 95% CI, 1.034–4.584; *p* = 0.041) and CPB time (OR for 1 h increase in CPB time, 4.194; 95% CI, 2.005–8.773; *p* < 0.000), even adjusted by age, sex, and BMI. Thus, preoperative ET-1 level and CPB time were identified as independent factors for the development of postoperative AKI.

An ROC curve was plotted to identify the optimal cutoff level of CPB time to detect AKI as shown in Figure 2. To generate the ROC curve, different CPB times were used to predict AKI with true positives on the vertical axis (sensitivity) and false positives (1 – specificity) on the horizontal axis. The area under the curve (AUC) for CPB time was 87%. Sensitivity and specificity were 90% and 70.2%, respectively. The optimal cutoff value was 178.5 min.

### 3.4. Preoperative ET-1 and GDF-15 Level as Short-Term Risks

We examined the relationships between STS score and preoperative GDF-15 and ET-1 levels and postoperative risks. As shown in Table 3, the preoperative GDF-15 level was correlated with ICU stay period (r = 0.191, *p* = 0.012) and postoperative admission days (r = 0.211, *p* = 0.006). The preoperative ET-1 level was also correlated with ICU stay period (r = 0.276, *p* = 0.001) and postoperative admission days (r = 0.247, *p* = 0.003). The STS score was positively correlated with ICU stay period (r = 0.351, *p <* 0.001) and postoperative admission days (r = 0.339, *p* < 0.001).

Furthermore, multivariable logistic regression analysis with 30-day mortality plus morbidity as the dependent variable was performed to identify independent factors (preoperative and intraoperative data), as shown in Table 5. In total patients undergoing HD, multivariable logistic regression analysis showed that the risk of 30-day mortality plus morbidity increased with increasing preoperative GDF-15 level at baseline (OR for 1 pg/mL increase in GDF-15, 1.001; 95% CI, 1.000–11.001; *p* = 0.030), CPB time (OR for 1 h increase in CPB time 1.947; 95% CI, 1.023–3.705; *p* = 0.042), and RCC transfusion (OR for 1 U increase in RCC transfusion 1.157; 95% CI, 1.041–1.286; *p* = 0.007), even adjusted for age, gender, and BMI (Table 5A). Similarly, the preoperative GDF-15 level, eGFR, CPB time, and RCC transfusion were independent factors to increase 30-day mortality plus morbidity (Table 5B) in non-HD patients.

## 4. Discussion

The major findings of the present study are as follows: 1) The preoperative GDF-15 and ET-1 levels correlated with STS score. The STS score and GDF-15 and ET-1 levels correlated with eGFR, BNP, Alb, and Hb. 2) The preoperative GDF-15 and ET-1 levels and the STS score correlated with the intraoperative data (urine volume) and postoperative outcomes (ICU stay period and postoperative admission days). 3) Patients with AKI had longer CPB time, and male patients with AKI had higher ET-1 levels than those without AKI. 4) In multivariable logistic regression analysis, the preoperative ET-1 level as well as CPB time was an independent determinant of AKI. 5) The preoperative GDF-15 level as well as RCC transfusion and CPB time was an independent determinant of 30-day mortality plus morbidity. These results suggest that preoperative GDF-15 and ET-1 levels as well as intraoperative factors such as CPB time may be helpful to identify short-term operative risk for patients undergoing cardiovascular surgery.

The STS score is widely used in cardiovascular surgery mortality risk assessment [1,2,32,33]. The present study showed that the STS score correlated with eGFR, BNP Alb, and Hb, reflecting organ dysfunction or nutritional status [34], and correlated with ICU stay period and postoperative admission days. However, the STS score was not significantly different between patients with and without AKI, and it was not an independent factor of AKI in our present study. In the present study, 23% of patients developed postoperative AKI after cardiovascular surgery. Patients with AKI had low BMI, long CPB time, high bleeding, high RCC, FFP, PC, and long postoperative intubation time. In multivariable logistic regression analysis using the intraoperative data, CPB time was a major independent determinant of AKI, which was consistent with previous papers [11,12]. Karim et al. [13] reported a positive association between increasing CPB time and incidence of AKI. They showed that CPB time >70 min increased the risk of AKI by an OR of 4.76, which is consistent with the present study showing an OR of 4.194 for 1 h increase in CPB time. The optimal cutoff value of CPB time obtained from the ROC curve was 178.5 min. Several mechanisms underlying the development of AKI during CPB have been proposed. Nephrotoxins, CPB-induced hemolysis, ischemic reperfusion injury, complexity of cardiovascular surgery, oxidant stress, and several factors (surgical trauma, embolism, contact between blood components and artificial surfaces of the circuit, cardiac arrest, allogeneic blood transfusion) may activate leukocytes, platelets, and vascular endothelial cells, which causes a systemic inflammatory response that affects the heart, brain, lungs, and kidneys [10,35,36,37,38]. Thus, it is very likely that a decrease of CPB time is required to prevent AKI development during cardiovascular surgery. However, in contrast with the previous reports [9,39], the incidence of AKI in females was less than in males in our study. The reasons for this discrepancy are unclear. Nevertheless, different types of cardiovascular surgery including CABG, valve diseases, and aortic disease and preoperative pathophysiological conditions, such as CKD, and coexisting cardiovascular risk factors may be complexly involved. However, Neugarten and Golestaneh [40] have reported a meta-analysis showing that female sex does not confer a greater risk of AKI and instead has a protective role. Therefore, further studies using a large number of patients are needed to clarify this question.

In this study, GDF-15 was strongly positively correlated with preoperative eGFR level, which is consistent with a previous paper [29]. The known predictors of postoperative AKI include pre-existing renal dysfunction [9]. However, preoperative creatinine, eGFR, and GDF-15 level were not significantly different between patients with and without AKI. In addition, neither GDF-15 nor eGFR was an independent factor for the onset of AKI. Recent studies have reported that increased preoperative GDF-15 levels can predict AKI risk after elective cardiovascular surgery [30], especially in patients with normal creatinine. However, in the present study, the preoperative GDF-15 level was not significantly different between patients with and without AKI. The reasons for this discrepancy are unclear. Similarly, there are several papers showing no significant association between preoperative creatinine level and postoperative AKI [22].

The present study shows that in multivariable logistic regression analysis, the preoperative ET-1 level as well as CPB time are independent determinants of AKI. This is consistent with previous papers showing that the preoperative ET-1 level predicts postoperative AKI as defined by the RIFLE (risk, injury, failure, loss of kidney function, and end-stage kidney disease) criteria in patients undergoing cardiovascular surgery [22]. They showed that preoperative ET-1 was predictive of postoperative AKI (*p* = 0.016) in a total of 105 patients undergoing cardiac surgery, who developed AKI (28%). Similarly, in the present study, AKI developed in 23% of patients. Patients with AKI had longer CPB time, and male patients with AKI had higher ET-1 levels than those without AKI. In multivariable logistic regression analysis, the preoperative ET-1 level as well as CPB time was an independent determinant of AKI, adjusted by age, sex, and BMI. However, in contrast with the previous study [22], the preoperative ET-1 level in males was significantly higher in patients with AKI than that in those without AKI, but not in females, probably due to a small number of female patients with AKI. Therefore, further studies are needed to determine the preoperative ET-1 level for predicting AKI.

Both preoperative GDF-15 and ET-1 levels correlated well with STS score, but the preoperative GDF-15 level and STS score were not significantly different between patients with and without AKI. Infusion of endothelin has been reported to vasoconstrict the afferent and efferent glomerular arterioles, raise internal vascular resistance, reduce renal blood, and effectively decrease the glomerular filtration rate [26,41]. Furthermore, it has been reported that postoperative AKI is associated with higher preoperative circulating ET-1 after pulmonary endarterectomy in patients with chronic thromboembolic pulmonary hypertension [42]. Bond et al. [43] also showed that systemic and pulmonary endothelin levels remain elevated for at least 24 h after CABG. As such, preoperative ET-1 level may function as an excellent biomarker for monitoring, preventing, and potentially treating AKI itself. In fact, Patel et al. [44] have shown that antagonism of the endothelin-1A receptor reversed the changes associated with endothelial dysfunction, regional tissue hypoxia, and proximal renal tubular epithelial cell stress elicited by post-CPB-induced AKI in a swine model and may represent a therapeutic target for the prevention of AKI. Further clinical studies using ET-1 antagonist are needed to clarify the involvement of ET-1 in postoperative AKI after cardiovascular surgery.

GDF-15 has been described as a potent marker of cardiovascular events (death, recurrent heart failure, recurrent myocardial infarction) in various populations, including patients with myocardial infarction or heart failure [45]. In the present study, the preoperative GDF-15 level was correlated with eGFR, BNP, Alb, and Hb, reflecting organ dysfunction or nutritional status [34], and postoperative outcomes, such as ICU stay period and postoperative admission days. In multivariable logistic regression analysis, the intraoperative factors (CPB time and RCC transfusion) were identified as independent factors to increase 30-day mortality plus morbidity. Additionally, the preoperative GDF-15 level was an independent determinant of 30-day mortality plus morbidity in patients with and without HD. This is compatible with a previous paper [46] showing that preoperative GDF-15 level is an independent predictor of postoperative 30-day mortality and morbidity in patients undergoing cardiovascular surgery. They also showed that preoperative GDF-15 level can further stratify patients beyond established risk scores, such as the EuroSCORE, and other cardiovascular risk markers such as NTproBNP or hsTNT. It remains unclear why GDF-15 may be an independent determinant of short-term risk. However, we have reported that muscle wasting, including sarcopenia, is a short-term prognostic factor for patients after cardiovascular surgery [47], and elevation of GDF-15 level reflects muscle wasting as well as renal dysfunction in preoperative cardiovascular surgery patients [29]. Thus, sarcopenia may be an underlying prognostic factor for the development of postoperative complications and short-term risk, where GDF-15 may be involved.

This study has several limitations. First, the study had a small number of cardiovascular surgery patients, and the patients underwent different types of cardiovascular surgery. Therefore, our findings are not necessarily applicable to the general population of cardiovascular surgery patients. Furthermore, most of the subjects had medical treatment. The use of drugs such as β-blockers, angiotensin converting enzyme inhibitor (ACE-I), and angiotensin II receptor blocker (ARB) might have affected the postoperative outcomes (AKI development, short-term mortality, and morbidity). In addition, we classified AKI only according to the creatinine criteria. Finally, the present study only investigated preoperative data (eGFR, STS score, ET-1, and GDF-15 level) and the intraoperative factors such as CPB time that can cause or be predictive of short-term operative risks in cardiovascular surgery, comparing them and correlating them with each other. In addition, considering the reclassification rate in a model that adds the effects of preoperative GDF-15 and ET-1 markers to the perioperative risk assessment of the STS score may be a good predictive marker of short-term operative risks. Thus, further studies using a large number of patients and detailed analysis including preoperative, intraoperative, and postoperative factors are required to clarify whether preoperative GDF-15 and ET-1 levels can be markers of short-term operative risks for patients undergoing cardiovascular surgery.

## 5. Conclusions

The present study provides evidence that preoperative GDF-15 and ET-1 levels as well as intraoperative factors such as CPB time may be helpful to identify short-term operative risk for patients undergoing cardiovascular surgery.

## Figures and Tables

**Figure 1 jcm-10-01960-f001:**
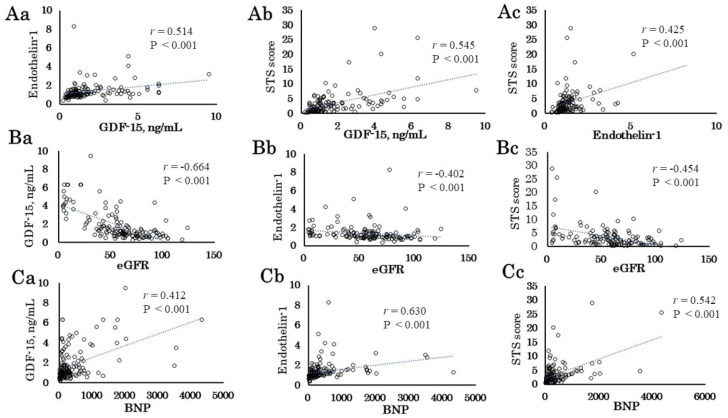
Correlations between clinical data and preoperative GDF-15, ET-1 level, and STS score. (**A**) Correlations between preoperative GDF-15 level, ET-1 level, and STS score. (**B**) Correlations between eGFR and preoperative GDF-15 level (**Ba**), ET-1 level (**Bb**), and STS score (**Bc**). (**C**) Correlations between BNP and preoperative GDF-15 level (**Ca**), ET-1 level (**Cb**), and STS score (**Cc**).

**Figure 2 jcm-10-01960-f002:**
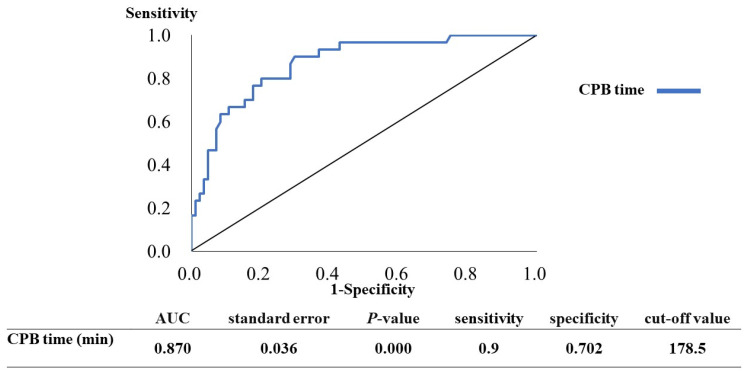
ROC curves to identify the optimal cutoff of CPB time for detecting AKI. To generate the ROC curves shown, CPB time cutoffs were used to predict AKI, with true positives plotted on the vertical axis (sensitivity) and false positives (1 – specificity) plotted on the horizontal axis.

**Table 1 jcm-10-01960-t001:** Patient characteristics.

Patients, Number	145
Risk factors, number	
Hypertension (HT), n (%)	113 (78)
Diabetes (DM), n (%)	48 (33)
Dyslipidemia (Dlp), n (%)	64 (44)
Smoking, n (%)	26 (18)
CKD, n (%)	76 (52)
Hemodialysis (HD), n (%)	19 (13)
Previous cardiac surgery, n (%)	3 (2)
NYHA classification	2.1 ± 0.9
Cardiovascular surgery, number	
CABG	34
AVR	17
MVR	4
Dual valve replacement	4
MV repair	24
CABG combined with valve procedure (AVR, MV repair, MVR)	9, 5, 2
AVR combined with MV repair	4
Aortic disease (AAR, TAR, HAR, etc.)	23
Others	19
Preoperative data	
Creatinine, mg/dL	1.6 ± 2.0
eGFR, mL/min/1.73 m^2^	56.6 ± 27.3
Hb, g/dL	12.5 ± 1.9
Alb, g/dL	3.9 ± 0.6
hsCRP, mg/L	0.9 ± 2.2
HbA1c, %	6.1 ± 0.9
BNP, pg/mL	422 ± 659
EF, %	56.7 ± 13.5
GDF-15, pg/mL	1851 ± 1638
ET-1, pg/mL	1.38 ± 0.92
STS score, %	3.6 ± 4.2
Operative data	
CPB time, h	2.19 ± 0.93
Operative bleeding, mL	677 ± 448
Operative urine output, mL	1555 ± 2598
Operative transfusion	
RCC, U	8.3 ± 7.3
FFP, U	7.6 ± 7.1
PC, U	12.6 ± 14.3
Postoperative intubation period, h	34.6 ± 79.3

The values shown are mean ± SD. NYHA, New York Heart Association; CABG, coronary artery bypass grafting; AVR, aortic valve replacement; MVR, mitral valve replacement; MV repair, mitral valve repair; TAP, tricuspid annuloplasty; AAR, ascending aorta replacement; HAR, hemiarch replacement; TAR, total arch replacement; STS, The Society of Thoracic Surgeons; GDF-15, growth differentiation factor 15; ET-1, endothelin-1; AKI, acute kidney injury; CPB time, cardiopulmonary bypass time; RCC, red blood cell concentrates; FFP, fresh frozen plasma; PC, platelet concentrates.

**Table 2 jcm-10-01960-t002:** Perioperative and postoperative complications.

Variable	Number	
AKI (male/female), n	25/4	
AKI (Stage)		
Stage 1	20	
Stage 2	5	
Stage 3	4	
New HD, n (%)	4 (3)	
Stroke, n (%)	4 (2)	
Prolonged ventilation (>48 h), n (%)	20 (13)	
DSWI, n (%)	5 (3)	
Re operation, n (%)	5 (3)	
ICU period, day	3.0 ± 4.4	
Postoperative admission days, days	33.2 ± 40.3	
30-day mortality, n (%)	3 (2)	NOMI 2 cases, stroke 1 case
30-day mortality + morbidity, n (%)	29 (20)	

The values shown are mean ± SD. HD, hemodialysis; DSWI, deep sternal wound infection; AKI, acute kidney injury; ICU, intensive care unit; NOMI, non-occlusive mesenteric ischemia.

**Table 3 jcm-10-01960-t003:** Relationships between GDF-15/ET-1/STS score and operative data.

	GDF-15	ET-1	STS Score
Operative time, h	0.168 (0.027 *)	0.067 (0.428)	0.163 (0.037 *)
CPB time, h	0.010 (0.900)	0.099 (0.239)	0.128 (0.102)
Arrest time, h	−0.096 (0.208)	0.026 (0.754)	0.059 (0.457)
Bleeding, mL	0.126 (0.101)	0.058 (0.498)	0.079 (0.316)
Transfusion			
RCC, U	0.322 (<0.001 ***)	0.102 (0.226)	0431 (<0.001 ***)
FFP, U	0.109 (0.156)	−0.020 (0.813)	0.196 (0.012 *)
PC, U	0.150 (0.051)	0.113 (0.181)	0.368 (<0.001 ***)
Urine output, mL	−0.418 (<0.001 ***)	−0.365 (<0.001 ***)	−0.390 (<0.001 ***)
Balance, mL	0.214 (0.005 **)	−0.124 (0.144)	0.138 (0.080)
Postoperative intubation period, h	0.113 (0.140)	0.190 (0.023 *)	0.162 (0.039 *)
ICU period, day	0.191 (0.012 *)	0.276 (0.001 **)	0.351 (<0.001 ***)
Postoperative admission, days	0.211 (0.006 **)	0.247 (0.003 **)	0.339 (<0.001 ***)

* <0.05 ** <0.01 *** <0.001. CPB time, cardiopulmonary bypass time; RCC, red blood cell concentrates; FFP, fresh frozen plasma; PC, platelet concentrates, ICU, intensive care unit.

**Table 4 jcm-10-01960-t004:** Multiple logistic regression analysis of presence or absence of AKI and clinical data.

A: Multiple logistic regression analysis of perioperative data				
	Dependent Variable: Presence or Absence of AKI			
Independent Variable	Odds ratio	95% Confidence Interval		*p*-Value
	Model 1/Model 2	Lower Limit	Upper Limit	
CPB time, h	3.617/3.404	1.807/1.544	7.240/7.053	<0.001 ***/0.002 **
Bleeding, mL	1.000/1.000	1.000/0.999	1.001/1.001	0.262/0.859
RCC transfusion, U	1.049/1.114	0.926/0.951	1.187/1.305	0.453/0.181
FFP transfusion, U	0.893/0.896	0.766/0.748	1.041/1.074	0147/0.236
PC transfusion, U	0.979/0.982	0.909/0.902	1.053/1.070	0.565/0.683
Postoperative intubation time, h	1.012/1.021	0.990/0.991	1.035/1.051	0.271/0.168
B: Multiple logistic regression analysis of preoperative and perioperative data				
	Dependent Variable: Presence or Absence of AKI			
Independent Variable	Odds ratio	95% Confidence Interval		* p *-Value
	Model 1/Model 2	Lower Limit	Upper Limit	
ET-1, pg/mL	1.926/2.300	1.105/1.144	3.357/4.624	0.021 */0.019 *
GDF-15, pg/mL	1.000/1.000	1.000/1.000	1.000/1.001	0.075/0.227
RCC transfusion, U	1.047/1.137	0.951/1.007	1.153/1.284	0.346/0.038 *
FFP transfusion, U	0.980/0.966	0.870/0.839	1.104/1.113	0.741/0.636
CPB time, h	4.115/4.113	2.103/1.892	8.051/8.943	<0.001 ***/<0.001 ***

A: Model 1, adjusted by CPB time, bleeding, RCC transfusion, FFP transfusion, PC transfusion, and postoperative intubation time. Model 2, adjusted by CPB time, bleeding, RCC transfusion, FFP transfusion, PC transfusion, postoperative intubation time, age, sex, and BMI. B: Model 1, adjusted by ET-1, GDF-15, RCC transfusion, FFP transfusion, and CPB time; Model 2, adjusted by ET-1, GDF-15, RCC transfusion, FFP transfusion, CPB time, age, sex, and BMI. * *p* < 0.05, ** *p* < 0.01, *** *p* < 0.001.

**Table 5 jcm-10-01960-t005:** Multiple logistic regression analysis of 30-day mortality + morbidity and clinical data.

A: Multiple logistic regression analysis of 30-day mortality + morbidity (total patients)				
	Dependent Variable: Dependent Variable: 30-Day Mortality + Morbidity			
Independent Variable	Odds ratio	95% Confidence Interval		*p*-Value
		Lower Limit	Upper Limit	
ET-1, pg/mL	1.105	0.463	2.634	0.822
GDF-15, ng/mL	1.001	1.000	1.001	0.030 *
eGFR, ml/min/1.73 m^2^	1.028	0.994	1.064	0.102
CPB time, h	1.947	1.023	3.705	0.042 *
RCC transfusion, U	1.157	1.041	1.286	0.007 **
B: Multiple logistic regression analysis of 30-day mortality + morbidity (except for dialysis patients)				
	Dependent Variable: Presence or Absence of AKI			
Independent Variable	Odds ratio	95% Confidence Interval		*p*-Value
		Lower Limit	Upper Limit	
ET-1, pg/mL	1.119	0.487	2.570	0.791
GDF-15, ng/mL	1.001	1.000	1.001	0.019 *
eGFR, ml/min/1.73 m^2^	1.031	1.002	1.062	0.037 *
CPB time, h	1.813	1.083	3.037	0.024 *
RCC transfusion, U	1.141	1.042	1.249	0.004 **

A: Model: adjusted by ET-1, GDF-15, eGFR, CPB time, RCC transfusion, age, sex, and BMI. B: Model: adjusted by ET-1, GDF-15, eGFR, CPB time, RCC transfusion, age, sex, and BMI. * *p* < 0.05, ** *p* < 0.01.

## Data Availability

The data presented in this study are available on request from the corresponding author. The data are not publicly available due to part of future research.

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
