# Peer review of "Preoperative Serum GDF-15, Endothelin-1 Levels, and Intraoperative Factors as Short-Term Operative Risks for Patients Undergoing Cardiovascular Surgery"

_jcm, 2021, doi:10.3390/jcm10091960_

Round 1

Reviewer 1 Report

I have read with interest the manuscript entitled "Preoperative serum GDF-15 and endothelin-1 levels as short-term operative risk factors for patients undergoing cardiovascular surgery" by Kato et al.. The authors evaluate the usage of preoperative serum GDF-15 and endothelin-1 levels in patients undergoing cardiovascular surgery.

However, I have some major concerns I would like to address: 

Introduction

  • In my opinion, the introduction should more clearly present the reason to perform the current study and its objectives.

Material and Methods

  • I would suggest clearly pointing out the design of the study. Is it a retrospective study (line 93) or a prospective study? Does it have a cross-sectional design (line 390)? Did the recruited patients undergo an emergency or a planed operation? That could have a huge impact on the results and their interpretation.  
  • Patients with different conditions undergoing different procedures (line 99-103) were all included in the study. However, they were all treated in a pooled fashion regarding post-op AKI. Were there differences regarding the endpoints among the different subpopulations? 
  • In my opinion, Table 1-2, as well as the baseline characteristics of the patients (line 94-98) should be presented in the results' section. 
  • I would suggest clearly stating the endpoints of the study and the time of the follow-up (7 days post-op?) in a clearer way in the manuscript. Moreover, the AKI classification refers solely to AKI 1. Were further AKI stages not differentiated? Could endothelin-1 and GDF-15 discriminate between the different stages? 
  • In Table 2: 4 patients have newly begun HD (meaning that AKI stage 3 criteria fulfilled), but only 3 had AKI Stage 3.
  • I have some concerns regarding the presented statistical Analysis: 
    • Are all the data normally distributed? If not, why did the authors present them as mean +/- SD and not as medians +/-IQR? 
    • I believe it would be better to clearly state when Mann-Whitney and when t-test were used. 
    • Why was Kolmogorov-Smirnov test was used, when Mann-Whitney U Test can be used for non-parametric data? 
    • When did the authors use Pearson and when Spearman correlation coefficients? 
    • What model (e.g. backwards, stepwise, etc.) was used for the multivariable logistic regression analysis? 
    • What power did the study have and what endpoint was it powered? 
    • Did the authors use an one- or a two-sided p-value? 
  • The majority of the patients had a pre-existing renal problem at the time of the surgery (Table 1, eGFR 56,6 ml/ml/1,73 m2 -> CKD Stadium G3a). However, in contrary to what the authors state in the introduction, neither female sex, nor pre-existing renal problem played a role in predicting post-op AKI in their cohort. This discrepancy should be further commented in the discussion. 

Results

  • In my opinion, it would be easier to highlight the most important correlations in the text and to present the rest in tables. Moreover, I do believe, that the main manuscript would be easier to follow, if some of the tables are presented as a supplement. That way it would be easier to focus on the main text.
  • Table 5: The number of males in AKI (-) cohort is 51, although in the last column is 52. Besides that, what do the numbers in bracket represent? P values should be also reported. 
  • Table 6: Are HD patients excluded? Did the authors adjust for surgery type and presence of CKD preoperatively? 
  • It might be of benefit, if the authors try to combine the two markers with STS score and maybe develop a more sensitive score.

Discussion

  • Line 292: + line 304: I would kindly disagree with the authors regarding "nutritional status". What data reflect that point? 
  • Line 341 - 349: I would kindly disagree with the authors at that point. Creatinine levels below 180 mg/dl are not considered to be normal. It would be possible more correct to assess and refer to eGFR rather than creatinine values. Pre-operative eGFR values of 45 and 31 reflect a CKD stage G3b (moderate-severe dysfunction)
  • I would suggest trying to answer the question, why endothelin-1 and GDF-15 can be used as risk markers. 

All in all, the idea is of interest, but - in my opinion - the current manuscript has "no straight storyline". I believe that a substantial revision is being needed. I would suggest stating a clear hypothesis with clear endpoints. The results' section should highlight the most important results supporting the hypothesis, while the rest of the data should be presented in the supplement. I would also advice the authors to present more data as figures (rather than tables) and expand their analysis by using ROC curves and Kaplan-meier analysis.

Author Response

Reply to Reviewer 1

We greatly appreciate your careful attention to our manuscript and especially your excellent suggestions for improving the clarity and correctness of the message. We have corrected the paper as per your suggestions, and consider the revised manuscript much improved.

Introduction

  • In my opinion, the introduction should more clearly present the reason to perform the current study and its objectives.

#) Answer: Thank you very much for your suggestions. We have carefully checked and revised the whole of the manuscript. And, we added the following sentence.

Page 2 line 62 Biomarkers can also provide insights into underlying mechanisms and lead to better understanding of the pathophysiology of complex disease states such as AKI. This enhanced understanding can then be integrated into disease management which can lead to better therapies and ultimately to improved patient outcomes.

Material and Methods

  • I would suggest clearly pointing out the design of the study. Is it a retrospective study (line 93) or a prospective study? Does it have a cross-sectional design (line 390)? Did the recruited patients undergo an emergency or a planned operation? That could have a huge impact on the results and their interpretation.

#) Answer: You are right. This study is a prospective design. We corrected it. The recruited patients have received a planned operation. We mentioned it in methods.

Page 2 line 99: We prospectively investigated the onset of AKI and a short-term risk after the operation.

Page 2 line 94: Patients undergoing planned cardiovascular surgery with cardiopulmonary bypass (CPB) during thoracotomy or median sternotomy at Dokkyo Medical Hospital from October 2015 to December 2020 were included in this study.

  • Patients with different conditions undergoing different procedures (line 99-103) were all included in the study. However, they were all treated in a pooled fashion regarding post-op AKI. Were there differences regarding the endpoints among the different subpopulations? 

#) Answer: Thank you very much for your comments. There were no particular differences regarding the endpoints among the different subpopulations.

  • In my opinion, Table 1-2, as well as the baseline characteristics of the patients (line 94-98) should be presented in the results' section. 

#) Answer: I moved them to results.

  • I would suggest clearly stating the endpoints of the study and the time of the follow-up (7 days post-op?) in a clearer way in the manuscript. Moreover, the AKI classification refers solely to AKI 1. Were further AKI stages not differentiated? Could endothelin-1 and GDF-15 discriminate between the different stages?

#) Thank you very much for your comments. The end point and time of the follow-up was one month after the operation. I mentioned it (Page 2 line 100). As shown in Table 2, the number of stage 2 is 5 patients, and the number of stage 3 is only 4 patients. Therefore, we have not obtained any significant differences among risk score and ET-1 and GDF-15 level in AKI stage. We commented it in results.

Page 7 line 245 On the other hand, as shown in Table 2, the number of stage 2 was 5 patients, and the number of stage 3 was 4 patients. We have not obtained the statistical difference between AKI stage and STS score, ET-1, and GDF-15 level (data not shown).

  • In Table 2: 4 patients have newly begun HD (meaning that AKI stage 3 criteria fulfilled), but only 3 had AKI Stage 3.

Answer: Sorry. I have a mistake, and I corrected it.

  • I have some concerns regarding the presented statistical analysis: 
    • Are all the data normally distributed? If not, why did the authors present them as mean +/- SD and not as medians +/-IQR? 

Answer: Thank you very much for your comments. No. some of the data are normally distributed, and some are not normally distributed. After the check, we presented all the data as mean +/- SD.

I believe it would be better to clearly state when Mann-Whitney and when t-test were used. 

#) Answer: Some of the data are normally distributed, and some are not normally distributed. After testing for normality (Kolmogorov-Smirnov), the comparison of means between groups were analyzed by a two-sided, unpaired Student’s t-test in the case of normally distributed parameters or by the Mann-Whitney-U-Test in the case of non-normally distributed parameters. I mentioned it in results (Page 4 line 157~)

    • Why was Kolmogorov-Smirnov test was used, when Mann-Whitney U Test can be used for non-parametric data? 

#) Answer: Thank you very much for your comments. Kolmogorov-Smirnov test was used to check if the data are normally distributed because the number of data are above 20-25.

    • When did the authors use Pearson and when Spearman correlation coefficients? 

#) Answer; Pearson was used when both data are normally distributed, and Spearman correlation coefficients was used when either of two data is not normally distributed.

    • What model (e.g. backwards, stepwise, etc.) was used for the multivariable logistic regression analysis? 
  • Clinical, laboratory and physical data should also be included in the analyses.

#) Answer: Thank you very much for your comments. To select independent factors in logistic regression analysis, differences in preoperative or perioperative factors with clinical outcome were examined using the U- and t-tests. The variable reduction method with all the factors that gave a significant difference was used to determine the independent factors in logistic regression analysis. In addition, ET-1 and GDF15, a biomarker investigated in this study, were considered as independent factors from the clinical aspect in this study. Using the independent factors obtained from the above results, logistic regression analysis was performed using the forced input method. Age, sex, and BMI were employed as covariates (Page 4 line 165).

    • What power did the study have and what endpoint was it powered?

#) Answer: Thank you very much for your comments. We have not calculated the power and sample size in this study. We have referred reference 18. They investigated 105 patients undergoing cardiac surgery (either coronary bypass surgery and/or valve surgery), and out of 105 patients, 29 patients (28%) developed AKI postoperatively. Therefore, we investigated total 126 patients without HD.

    • Did the authors use an one- or a two-sided p-value? 

#) Answer; Thank you very much for your comments. We used a two-sided P-value.

  • The majority of the patients had a pre-existing renal problem at the time of the surgery (Table 1, eGFR 56,6 ml/ml/1,73 m2 -> CKD Stadium G3a). However, in contrary to what the authors state in the introduction, neither female sex, nor pre-existing renal problem played a role in predicting post-op AKI in their cohort. This discrepancy should be further commented in the discussion. 

#) Answer: Thank you very much for your comments. We are very surprised at the unexpected results. However, several papers have reported the results. We have commented it in discussion.

Page 10 line 365~ However, in contrast with the previous reports [8,39], the incidence of AKI in females was less than in males in our study. The reasons for this discrepancy are unclear. But, different types of cardiovascular surgery including CABG, valve diseases, and aortic disease and preoperative pathophysiological conditions such as CKD, and coexisting cardiovascular risk factors may be complexly involved. However, Neugarten and Golestaneh [40] have reported a meta-analysis showing that female sex does not confer a greater risk of AKI and instead has a protective role. Therefore, further studies using a large number of patients are needed to clarify this question.

Page 11 line 362~ Thus, the present study shows that preoperative renal dysfunction was not a significant independent factor of AKI in our present study. The known predictors of postoperative AKI include pre-existing renal dysfunction [8,9]. The reasons for this discrepancy are unclear. Similarly, there are several papers showing no significant association between pre-operative creatinine level and postoperative AKI [18].

Results

  • In my opinion, it would be easier to highlight the most important correlations in the text and to present the rest in tables. Moreover, I do believe, that the main manuscript would be easier to follow, if some of the tables are presented as a supplement. That way it would be easier to focus on the main text.

#) Thank you very much for your suggestion. We removed some tables into the supplement files.

  • Table 5: The number of males in AKI (-) cohort is 51, although in the last column is 52. Besides that, what do the numbers in bracket represent? P values should be also reported. 

#) Sorry. We have corrected it, and we added the P-value.

Table 6: Are HD patients excluded? Did the authors adjust for surgery type and presence of CKD preoperatively? 

#) Thank you very much for your comments. Of course, I exclude HD patients. But, unfortunately, we have not adjusted surgery types, because a small number of patients had developed AKI. As your suggestion, we have not significant difference of AKI development among CKD stage.

It might be of benefit, if the authors try to combine the two markers with STS score and maybe develop a more sensitive score.

#) Answer: Thank you very much for your suggestion. I also hoped that the combination of the two markers with STS score maybe develop a more sensitive score. However, unfortunately we could not obtain such data probably due to a small number of patients with AKI, and the further studies using a large number of patients are required to clarify it.

Discussion

  • Line 292: + line 304: I would kindly disagree with the authors regarding "nutritional status". What data reflect that point? 

#) Answer: You are right. We deleted it.

  • Line 341 - 349: I would kindly disagree with the authors at that point. Creatinine levels below 180 mg/dl are not considered to be normal. It would be possible more correct to assess and refer to eGFR rather than creatinine values. Pre-operative eGFR values of 45 and 31 reflect a CKD stage G3b (moderate-severe dysfunction)

#) Answer; You are right. We corrected it.

This is consistent with previous papers showing that the preoperative ET-1 level predicts postoperative AKI as defined by the RIFLE (Risk, Injury, Failure, Loss of kidney function, and End-stage kidney disease) criteria in patients undergoing cardiovascular surgery [18]. They showed that preoperative ET-1 was predictive of post-operative AKI (P=0.016) in total 105 patients undergoing cardiac surgery, who developed AKI (28%). Similarly, in the present study, AKI developed in 23% of patients.

  • I would suggest trying to answer the question, why endothelin-1 and GDF-15 can be used as risk markers.

#) Answer: Thank you very much for your suggestion. In this study, preoperative ET-1 level and CPB time were identified as independent factors for the development of postoperative AKI. In addition, the preoperative GDF-15 level was an independent determinant of 30-day mortality plus morbidity. These results suggest that preoperative ET-1 and GDF-15 levels may be a marker for short-term operative risk, including AKI, for patients undergoing cardiovascular surgery.

All in all, the idea is of interest, but - in my opinion - the current manuscript has "no straight storyline". I believe that a substantial revision is being needed. I would suggest stating a clear hypothesis with clear endpoints. The results' section should highlight the most important results supporting the hypothesis, while the rest of the data should be presented in the supplement. I would also advice the authors to present more data as figures (rather than tables) and expand their analysis by using ROC curves and Kaplan-meier analysis.

#) Answer: Thank you very much for your comments. I absolutely agree with your opinion. I have new data of ROC curve (Figure 2).

The methodology used is appropriate, the interpretation of the results is solid and the study adds value to the medical field of application.

#) Answer: Thank you very much for your suggestion. We mentioned about the clinical value of the present study in discussion.

Reviewer 2 Report

In this work entitled “Preoperative Serum Gdf-15 and Endothelin-1 Levels as Short-Term Operative Risk Factors for Patients Undergoing Cardiovascular Surgery”, Kato T et al investigated the clinical usefulness of preoperative serum GDF-15 and ET-1 levels as markers for short-term operative risks, including AKI, for patients undergoing cardiovascular surgery.

The methodology used is appropriate, the interpretation of the results is solid and the study adds value to the medical field of application.

Major changes

None

Minor changes

  1. A minor revision of the English grammar used would benefit the article greatly
  2. In the introduction section, acute kidney injury (AKI) is spelled out twice (Page 1, line 42 and line 45)
  3. The authors state in the introduction that: “the usefulness of the risk score [STS] is not high enough yet.” A short sentence explaining why would be appreciated.
  4. Instead of “The Regional Ethics Committee”, use the full name of the committee and, if possible, an identifier.
  5. In the methods section, the variables BMI, HT, DM…are spelled twice (in the “participants” subsection and the , “items to consider” subsection)
  6. In the results section, “The clinical characteristics of the study patients are shown in Table 1.” Is not necessary as it was already presented in the methods (Participants) section.
  7. The authors could compare the net reclassification rate of [STS vs GDF-15 and/or ET-1] vs [STS+GDF15 and/or ET-1] using a Hosmer-Lemeshow test.

Author Response

Reply to Reviewer 2

We greatly appreciate your careful attention to our manuscript and especially your excellent suggestions for improving the clarity and correctness of the message. We have corrected the paper as per your suggestions, and consider the revised manuscript much improved.

Minor changes

  1. A minor revision of the English grammar used would benefit the article greatly.

#) Answer: Thank you very much for your suggestions. We have checked and revised the whole of the manuscript.

  1. In the introduction section, acute kidney injury (AKI) is spelled out twice (Page 1, line 42 and line 45)

#) Answer: I deleted one.

  1. The authors state in the introduction that: “the usefulness of the risk score [STS] is not high enough yet.” A short sentence explaining why would be appreciated.

#) Answer: The risk score (STS) has been wildly used to evaluate risk score. We have changed it as follows.

Page 2 line 53 However, the STS score is not sufficient in terms of predicting mild to moderate postoperative AKI that does not lead to the introduction of dialysis.

  1. Instead of “The Regional Ethics Committee”, use the full name of the committee and, if possible, an identifier.

#) Answer: The Regional Ethics Committee of Dokkyo Medical University has approved the study protocol

  1. In the methods section, the variables BMI, HT, DM…are spelled twice (in the “participants” subsection and the “items to consider” subsection)

#) Answer: I corrected it.

  1. In the results section, “The clinical characteristics of the study patients are shown in Table 1.” Is not necessary as it was already presented in the methods (Participants) section.

#) Answer: I deleted it.

  1. The authors could compare the net reclassification rate of [STS vs GDF-15 and/or ET-1] vs [STS+GDF15 and/or ET-1] using a Hosmer-Lemeshow test.

#) Answer: A: Thank you very much for your useful suggestion. As you pointed out, considering the reclassification rate in a model that adds the effects of GDF-15 and ET-1 markers to the perioperative risk assessment of the STS score may be a good predictive model. However, the main purpose of the present study is to examine the effects of serum markers, endothelin-1 and GDF-15, and intraoperative factors. Therefore, in future ongoing studies, we will examine this regression model using STS scores and serum markers to further improve accuracy by using a large number of patients.

Round 2

Reviewer 1 Report

I would like to thank the authors for considering my comments and editing their manuscript. Although I do find that it has been improved, my greatest concern has not been adequately answered. 

The present manuscript does not focus to answer the hypothesis made “to determine whether preoperative ET1 and GDF15 levels can be used as preoperative serum bio markers to help identify patients at AKI risk and operative risk after cardiovascular surgery.” I believe that the structure of the manuscript should be substantially revised and focus to answer the presented question:“Does preoperative ET1 and GDF15 predict postoperative AKI?”

Instead it presents different factors that can cause or be predictive of AKI in cardiovascular surgery comparing them and correlating them with each other. In the current version of the manuscript, the authors focus to answer a different question: “what factors predict or associate with post-operative AKI?” 

Author Response

Reply to Reviewer 2

We greatly appreciate your careful attention to our manuscript and especially your excellent suggestions for improving the clarity and correctness of the message. We have corrected the paper as per your suggestions, and consider the revised manuscript much improved.

I would like to thank the authors for considering my comments and editing their manuscript. Although I do find that it has been improved, my greatest concern has not been adequately answered.

The present manuscript does not focus to answer the hypothesis made “to determine whether preoperative ET1 and GDF15 levels can be used as preoperative serum bio markers to help identify patients at AKI risk and operative risk after cardiovascular surgery.” I believe that the structure of the manuscript should be substantially revised and focus to answer the presented question “Does preoperative ET1 and GDF15 predict postoperative AKI?”

Instead, it presents different factors that can cause or be predictive of AKI in cardiovascular surgery comparing them and correlating them with each other. In the current version of the manuscript, the authors focus to answer a different question: “what factors predict or associate with post-operative AKI?”

#) Answer: Thank you very much for your suggestion. I absolutely agree with your suggestions. The main purpose of the present study is to investigate what factors predict or associate with post-operative AKI. Therefore, we have corrected the revised paper as follows. 

Major points

We changed the title. I added the intraoperative factors.

Preoperative Serum GDF-15, Endothelin-1 Levels and Intraoperative Factors as Short-Term Operative Risks for Patients Undergoing Cardiovascular Surgery

Page 1line 19I

We investigated the clinical usefulness of preoperative serum GDF-15 and ET-1 levels as markers for short-term operative risks including acute kidney injury (AKI) for patients undergoing cardiovascular surgery. →

 We investigated the relationships between preoperative serum GDF-15, ET-1 levels and intraoperative factors and short-term operative risks including acute kidney injury (AKI) for patients undergoing cardiovascular surgery.

Page 1 line 33

 Conclusion: Elevated preoperative GDF-15 and ET-1 levels can be markers of short-term operative risk for patients undergoing cardiovascular surgery.→

 Conclusion: Preoperative GDF-15, and ET-1 levels and intraoperative factors such as CPB time may be helpful to identify short-term operative risk for patients undergoing cardiovascular surgery.

Page 2 line 90: Therefore, to determine what factors predict or associate with operative risks including AKI, we investigated the relationships between preoperative serum GDF-15, ET-1 levels and intraoperative factors and short-term operative risks for patients undergoing cardiovascular surgery.

Page 10 line 319: 5) The preoperative GDF-15 level as well as RCC transfusion and CPB time was an independent determinant of 30-day mortality plus morbidity. These results suggest that preoperative GDF-15, and ET-1 levels and intraoperative factors such as CPB time may be helpful to identify short-term operative risk for patients undergoing cardio-vascular surgery.

Page 11 line 380: However, in contrast with the previous study [18], the preoperative ET-1 level in males was significantly higher in patients with AKI than that in those without AKI, but not in females, probably due to a small number of female patients with AKI. Therefore, the further studies are needed to determine the preoperative ET-1 level for predicting AKI.

Page 12 line 428: Finally, the present study investigated different factors that can cause or be predictive of AKI in cardiovascular surgery comparing them and correlating them with each other. And, we found that the intraoperative factors such as CPB time can predict or associate with post-operative AKI, but, further studies using a large number of patients, especially females, and detailed AKI criteria are required to clarify whether preoperative GDF-15 and ET-1 levels can be markers of short-term operative risks for patients undergoing cardiovascular surgery.

Page 12 line 435: The present study provides evidence that preoperative GDF-15, and ET-1 levels and intraoperative factors such as CPB time may be helpful to identify short-term operative risk for patients undergoing cardiovascular surgery.
